# Deltopectoral Flap in Head and Neck Reconstruction

**DOI:** 10.3390/medicina60101615

**Published:** 2024-10-02

**Authors:** Giancarlo Pecorari, Francesco Bissattini, Dario Gamba, Claudia Pizzo, Gian Marco Motatto, Giuseppe Riva

**Affiliations:** Division of Otorhinolaryngology, Department of Surgical Sciences, University of Turin, Via Genova 3, 10126 Turin, Italy; giancarlo.pecorari@unito.it (G.P.); francesco.bissattini@unito.it (F.B.); dario.gamba@unito.it (D.G.); claudia.pizzo@unito.it (C.P.); gian.motatto@gmail.com (G.M.M.)

**Keywords:** deltopectoral flap, head and neck cancer, reconstruction, head and neck surgery

## Abstract

*Background and Objectives*: The deltopectoral (DP) flap represents a reconstructive option for the head and neck. It is a fasciocutaneous flap raised from the anterior chest wall below the clavicle. Its role partially declined with the arise of free flaps. However, it still remains a valid option in patients that could not undergo a reconstruction with free flaps. The aim of this retrospective study was to evaluate the role of the DP flap in head and neck reconstruction, with a focus on surgical outcomes and complications. *Materials and Methods*: Thirty-one patients who underwent head and neck reconstruction with DP flap were included in the study. The delayed technique was not used in any procedure to harvest the flap beyond the deltopectoral groove. The patients’ characteristics, the recipient site, the closure of the donor site, and the flap-related complications were recorded. *Results*: The median time to autonomization of DP flap was 23 days. Postoperative complications were observed in 10 subjects (32.3%). A partial necrosis was seen in five patients (16.1%), while a complete necrosis developed only in one case (3.2%). The diabetes mellitus was the only parameter associated with postoperative complications. In particular, the percentage of necrosis in subjects with or without diabetes was 66.7% and 8.0%, respectively. *Conclusions*: The DP has a wide range of applications in head and neck reconstruction, and a low complication rate can be observed. The delayed technique does not necessarily need to be applied, and the flap can be extended beyond the deltopectoral groove when necessary. However, patients with diabetes mellitus have a higher risk of postoperative necrosis of the distal portion of the flap.

## 1. Introduction

Head and neck cancer represents a significant portion of global cancer incidence, accounting for more than 660,000 new cases and 325,000 deaths annually [1]. These tumors pose substantial challenges due to the complex anatomy and the critical functions associated with this region. Surgical resection, often the main treatment modality, results in complex defects that require meticulous reconstruction to restore both function and aesthetics. Over the years, various reconstructive techniques have been developed to address these challenges, ranging from local tissue flaps to free tissue transfers. Among these, the deltopectoral (DP) flap, first described by Bakamjian in 1965, has continued to play a crucial role in head and neck reconstruction despite the advent of the pectoralis major flap and more advanced microsurgical techniques [2].

The DP flap is a fasciocutaneous flap raised from the anterior chest wall below the clavicle. The flap is delimited medially by the sternum and laterally by the deltopectoral groove, although in some cases it can be extended 3–4 cm laterally in the acromioclavicular area. Superiorly it is delimited by the clavicle and inferiorly by the inferior margin of the fourth rib. Vascularization is based on perforators of the internal thoracic artery (ITA). The ITA originates from the subclavian artery and supplies the anterior chest wall and the breast. The blood vessels of the flap run on a plane superficial to the fascia that covers the pectoralis major and deltoid muscles; hence the importance of lifting these flaps on a deep plane with respect to the fasciae of the deltoid and pectoralis major muscles. The distal portion of the flap, which lies over the deltoid muscle, is supplied by the deltoid perforators from the thoracoacromial trunk and by the anterior circumflex artery and vein [3,4]. Therefore, the distal part represents a random flap beyond the axial blood supply of the IMAs. Due to the less reliable blood supply in this area, a delayed technique was proposed for successful harvesting [5,6]. The sensory supply to the deltopectoral skin is derived from the supraclavicular nerves of C3 and C4 and the anterolateral intercostal nerves of T2, T3, and T4. The ability to keep the sensory system intact depends greatly on whether a radical neck dissection is performed [7].

The DP flap is highly regarded for its reliability, straightforward dissection, and strong blood supply [8]. It offers a substantial area of well-vascularized tissue ideal for covering large defects, particularly in the head and neck region. Being a pedicled flap, it enables single-stage reconstruction, which reduces operative time and eliminates the need for microvascular anastomosis—an advantage for patients with comorbidities or in environments where microsurgical expertise is limited [9]. Furthermore, the donor site of the DP flap is generally located outside the field of irradiation, ensuring that the tissue used is healthy and well vascularized. Its positioning also results in a scar that is easily concealed, making it a cosmetically acceptable choice for patients concerned with aesthetic outcomes. If needed, a bilobed flap or bilateral flaps can be used, and the DP flap can be seamlessly combined with other pedicled or free flaps [10,11].

Although the DP flap is quite versatile, it has some drawbacks. In particular, it has a limited rotational arc, and, if a skin graft is required, the donor site may have an unattractive appearance, particularly in women, where it can lead to breast asymmetry. Additionally, in male patients, the flap may contain hair, which can negatively impact the aesthetic outcome. Complication rates for the DP flap have been reported to range from 0% (0 complications out of 16 DP flaps, 0 complications on the receiving site out of 24 DP flaps) to 51.4% (349 complications out of 678 DP flaps), with a complete flap failure rate ranging from 0% to 26% (out of 103 DP flaps) [12,13,14,15]. The success of the tissue transfer was affected by the length of the DP flap [14,16].

The recent literature has revisited the DP flap, exploring its potential roles in modern reconstructive surgery [17]. Some authors highlighted its utility in cases where free flaps are contraindicated or in situations where a simpler, less-time-consuming procedure is desirable. Additionally, modifications in the design and harvesting techniques of the DP flap have been proposed to enhance its versatility and outcomes, making it more adaptable to a wider range of defects [5,13].

Despite these developments, the role of the DP flap in contemporary head and neck reconstruction remains underexplored, particularly in comparison to the extensive body of literature on free flaps and the pectoralis major flap. Given the enduring relevance of the DP flap, especially in selecting patient populations and clinical contexts, there is a need for comprehensive studies that evaluate its efficacy, complications, and long-term outcomes.

This retrospective study aimed to evaluate the role of the DP flap in head and neck reconstruction, with a focus on surgical outcomes and complications. By analyzing a series of cases from our institution, we sought to provide insights into the contemporary applications of the DP flap and to assess its viability as a reconstructive option in the current surgical landscape.

## 2. Materials and Methods

Patients who underwent head and neck reconstruction with DP flap between 2017 and 2023 at our department were included in the study. Exclusion criteria were the following: age < 18 years, DP flap used for reconstruction of other areas. All procedures were in accordance with the ethical standards of the institutional research committee and with the 1964 Helsinki Declaration and its later amendments or comparable ethical standards. Written informed consent was obtained in every case. Institutional Review Board (A.O.U. Città della Salute e della Scienza di Torino—A.O. Ordine Mauriziano—A.S.L. Città di Torino) approval was obtained.

The DP flaps were harvested with the method described by Bakamjian, incorporating the second, third, and fourth intercostal perforating branches of the internal mammary artery [2]. The delayed technique was not used in any procedure to harvest the flap beyond the deltopectoral groove (all the cases). The flap was raised from 2 cm lateral to the sternum (to preserve the pedicle vessels) in the infra-clavicular area. It was raised in a subfascial plane from distal to proximal, including the fascia into the flap. Then, the DP flap was rotated to reconstruct the recipient site.

Three approaches were performed for the closure of the donor site: (1) wide undermining and primary closure (used in all the cases where it was feasible); (2) split-thickness skin autograft immediately after DP flap harvesting; (3) use of skin substitutes (bilayer Integra^®^ Dermal Regeneration Template, Integra Life Sciences, Plainsboro, NJ, USA) concomitant to DP flap harvesting and subsequent thin epidermal autograft concomitant to flap autonomization.

The Integra template is constituted by two layers: a thin outer layer of silicone and a thick inner matrix layer of pure bovine collagen and glycosaminoglycan derived from shark cartilage. It is pliable and soft, facilitating range of motion of the shoulder (donor site). The silicon layer provides a mechanical barrier and controls fluid loss. The matrix layer promotes cellular growth and collagen synthesis [18]. After 3 weeks, the silicon layer is removed and a thin epidermal graft can be applied over the neodermis.

The patients’ characteristics, the recipient site, the closure of the donor site, and the flap-related complications were recorded. In particular, the following postoperative complications were analyzed: partial and complete necrosis, wound dehiscence at the recipient site, bleeding, and wound infection.

All statistical analyses were carried out using the Statistical Package for Social Sciences, version 26.0. The Kolmogorov–Smirnov test demonstrated a non-Gaussian distribution of variables, so nonparametric tests were used. A descriptive analysis of all data was performed, and they were reported as medians and interquartile range (IQR), or percentages. The Mann–Whitney U-test was used for comparison between two independent groups (subjects with or without complications). The chi-squared or the Fisher exact test was used for categorical variables. A *p* < 0.05 was considered statistically significant.

## 3. Results

Thirty-one patients were included in the study. The median age at deltopectoral harvesting was 72 years (IQR 23 years), with a predominance of older subjects (Figure 1). Clinical characteristics are reported in Table 1.

The flap was used for nononcological reasons in one patient (3.2%) who had a traumatic lesion of the cervical esophagus (23 years old). Table 2 highlights the demolitive procedures performed before reconstruction. In seven cases (22.6%), the DP flap was used to close a pharyngocutaneous fistula after total laryngectomy. The median time to autonomization of DP flap was 23 days (IQR 9 days). Adjuvant therapy was performed in eight patients after the flap autonomization (chemoradiotherapy in three cases and radiotherapy in five cases). The recipient sites are shown in Figure 2.

The donor site was closed with primary closure in 9 subjects (29.0%), split-thickness skin autograft immediately after DP flap harvesting in 7 cases (22.6%), and skin substitutes (bilayer Integra^®^ Dermal Regeneration Template, Integra Life Sciences, Plainsboro, NJ, USA) concomitant to DP flap harvesting and subsequent thin epidermal autograft concomitant to flap autonomization in 15 patients (48.4%) (Figure 3). In the latter case, the epidermal autograft was harvested from the pedicle of the DP flap before removing it.

Postoperative complications were observed in 10 subjects (32.3%). The loss of the entire portion of the DP flap used for reconstruction was considered a complete necrosis. A partial necrosis was seen in five patients (16.1%), while a complete necrosis developed only in one case (3.2%). If a dehiscence of the flap at the recipient site occurred when it was used for the closure of a pharyngocutaneous fistula, the latter may reappear. Therefore, the new fistula was considered a postoperative complication (Figure 4).

The statistical analyses showed that diabetes mellitus was the only parameter associated with postoperative complications (Table 3). In particular, the percentage of necrosis in subjects with or without diabetes was 66.7% and 8.0%, respectively (*p* = 0.006).

The median hospital stay was 42 days (IQR 23 days). It should be considered that seven patients underwent a DP flap harvesting for pharyngocutaneous fistula after total laryngectomy. Therefore, such patients increased the median hospital stay. The appearance of postoperative complications did not significantly increase the hospital stay (*p* = 0.519). Indeed, most of the complications (partial necrosis, dehiscence without fistula, bleeding, infection) were treated with conservative therapies and/or dressings. A patient with postoperative dehiscence and a small fistula was treated by refreshing the borders of the wound dehiscence and performing a new suture in local anesthesia. On the contrary, a patient with a wide fistula needed a new flap (pectoralis major flap).

## 4. Discussion

The medially-based DP flap, described by Bakamjian in 1965, was first used for pharyngoesophageal reconstruction [2]. Then, it was widely used for head and neck reconstruction of skin and oral defects. In particular, the DP flap can be used for external defects of the pre- and retroauricular region, the cheek, the mentum, and the neck [14]. The blood supply is based on three different angiosomes: the first is based on IMA perforators, the second on perforators from the thoracoacromial trunk, and the third on perforators from the anterior circumflex artery [4]. Therefore, the distal portion of the DP flap has a variable reliability, and delayed techniques have been proposed to increase the viability of harvesting the DP flap beyond the deltopectoral groove [5]. However, the delayed technique needs one more surgical procedure that in some patients may not be advisable. Our study evaluated the surgical outcomes of head and neck reconstruction with a DP flap that was harvested beyond the deltopectoral groove without a delayed technique.

Since the 2000s, studies on the DP flap have increased in the literature. Unlike the many studies reported until 1990 that described DP flaps prepared with standard surgical techniques, recent analyses have focused on innovative or combined techniques. For example, authors described the pre-expanded DP flap, the use of it to close the donor site of a pectoralis major flap and reduce skin tension, the use of it in a single surgical procedure, or the combined use of the DP and pectoralis major flaps to reconstruct tissue loss in the head and neck region (PMID: 35712439) [11,19,20,21]. The novelty of our study was the analysis and identification of clinical factors that could contribute to the onset of postoperative complications in a standard DP flap.

The average age found in our case series was similar to that reported in the literature concerning the DP flap. Most of our cases involved head and neck reconstruction in an oncological setting, with patients having an age comparable to that with the higher incidence of these tumors (50–80 years) [1]. Similarly, in our study, most subjects were male, as is the case with tumors in the cervicofacial region [22]. Regarding smoking habits, two-thirds of our patients had never smoked. Regarding the remaining one-third, half were former smokers, and the other half were active smokers. Concerning alcohol consumption, only one-third of the patients were active drinkers. Globally, in terms of lifestyle habits, our sample aligned with the characteristics of patients diagnosed with head and neck cancer.

In our case series, only one patient was not affected by an oncological disease, but was involved in an accident that resulted in a rupture of the cervical esophagus following the trauma. The remaining 30 subjects were cancer patients, among whom, in 23 cases, the flap was used to fill the defect resulting from tumor resection, while in 7 cases, the DP flap was used to repair and close a pharyngocutaneous fistula that developed as a complication of a total laryngectomy. Although numerous cases of reconstruction with a DP flap following trauma have been reported in the literature [15,23,24], most of the cases where this flap has been used were oncological pathologies. In such cases, the defect resulting from tumor removal could be extensive, needing the use of the flap, or the repair of postoperative fistulas required tissue with sufficient thickness [14].

The recipient site varied and was evenly distributed throughout the head and neck region. Specifically, 26% of the flaps were used to reconstruct defects in the cheek area, another 26% for the chin and lower lip, 23% for the preauricular or retroauricular area, and the remaining cases for the neck and hypopharynx/cervical esophagus [14]. This demonstrates the great versatility of this flap, which, since the 1960s, has been used to reconstruct various sites within the head and neck region.

In the literature, the average time for the flap to become independent was 21 days, ranging from 3 to 6 weeks [25]. This time interval to autonomization of the DP flap aligned with the median of 23 days observed in our patients. Considering that all the flaps in our study were extended beyond the deltopectoral groove, it can be inferred that this surgical extension did not affect the time to autonomization in our case series.

Once the flap has undergone an autonomization procedure, according to clinical condition and disease staging, the patient could undergo adjuvant therapy, such as radiation therapy, chemotherapy, or concomitant chemoradiotherapy. In our series, none of the eight patients who underwent adjuvant therapy developed late flap complications after radiotherapy.

Three approaches were used in our case series for the closure of the donor site: (1) primary closure, (2) split-thickness skin autograft immediately after DP flap harvesting, and (3) use of skin substitutes (bilayer Integra^®^ Dermal Regeneration Template) concomitant to DP flap harvesting and subsequent thin epidermal autograft concomitant to flap autonomization. In the latter case, the epidermal autograft was harvested from the pedicle of the DP flap before removing it. About half of the donor sites were closed by means of dermal substitutes and an epidermal autograft harvested from the pedicle of the DP flap (these were the most recent cases). Such a procedure could avoid the morbidity of another site, usually the tight, used as donor of the epidermal graft. Indeed, in our patients, the donor area adjacent to the pedicle insertion site generally underwent a primary closure, whereas it was often impossible to perform a primary closure of the deltoid area. In the latter, the bilayer Integra^®^ Template was positioned. Since the medial portion of the pedicle was not necessary to close the donor site in such cases, it became the source of the epidermal autograft.

Postoperative complications occurred in 32% of patients, in agreement with those reported in the literature (Table 4), which ranged from 0% in some studies to about 50% in others with a significantly higher number of patients [12,13,14].

The most frequently observed complications in our study were partial necrosis and wound dehiscence. While the literature reported an incidence of up to 27% for partial flap necrosis [24], with our results that fall within the range of the literature, wound dehiscence was reported to occur in up to 21% of cases [25]. The higher percentage of dehiscence (21%) was reported in DP flaps harvested beyond the deltopectoral groove. Considering studies with less extended flaps, the higher rate of dehiscence was 6.9% [11]. Therefore, the fact that wound dehiscence occurred in 16% of cases in our study could be due to the surgical extension of the flap beyond the deltopectoral groove. Indeed, the distal portion of the DP flap may have suffered from excessive distance from the vascular pedicle, affecting the distal suture but not the integrity of the flap itself, thereby ensuring tissue survival. Moreover, in our case series, wound dehiscence was not related to a previous head and neck radiotherapy. On the other hand, a complete flap necrosis was observed only in one patient, whereas the literature reported this complication in up to 26% of cases [15], demonstrating that extending the flap beyond the deltopectoral groove without a delayed technique seemed not to negatively impact on flap survival. The occurrence of a fistula, reported in up to 18% of cases in the literature [26], was observed in only 6% of our subjects. Also, in such case, our complication rate was lower than those reported in the literature. Only one patient in our study experienced postoperative bleeding, which usually occurred in about 1% of patients reconstructed with a DP flap [14]. The last complication observed in our case series was a postoperative wound infection of the donor site in two patients. The fact that this complication occurred in up to 27% of cases in other studies [14,26] highlighted that an adequate postoperative management allowed us to maintain a sufficiently sterile environment during daily wound dressings. Thus, it can be noted that for all complications, except for dehiscence, our outcomes were better or were in agreement with the literature. This suggests that the extension of the flap beyond the deltopectoral groove without a delayed technique does not increase the risk of postoperative complications, except for wound dehiscence.

A retrospective study analyzed 34 subjects who underwent reconstruction with a DP flap extended beyond the deltopectoral groove, and the authors found a wound dehiscence in 21% of patients, emphasizing how the rate of this complication may increase when the flap was extended beyond the deltopectoral groove [25]. Another study compared 15 conventional DP flaps with 17 flaps extended beyond the deltopectoral groove and did not find any statistically significant differences between the two groups in terms of the incidence of postoperative complications [27]. An additional study analyzed 86 DP flaps and reported that most of these flaps were harvested extending beyond the mediolateral line of the shoulder without a delayed technique. Although the risk of complications was higher, no statistically significant difference was observed [15].

Out of 31 patients, only 5 did not have any comorbidities, while the remaining 26 subjects had at least one other disease. The most common comorbidity was systemic hypertension, followed by diabetes mellitus. This initial descriptive analysis showed that our study analyzed a sample of patients with medical histories that perfectly reflected those of individuals undergoing reconstruction with a DP flap, as reported in the literature [14].

The results of our analyses revealed a statistically significant association between the occurrence of partial or complete flap necrosis and the presence of diabetes mellitus. Necrosis in diabetic patients occurred in 67% of cases, whereas in nondiabetic patients, it occurred in only 8% of subjects. The reason for this correlation may lie in the fact that diabetic patients have microvascular damage, which, in the case of a DP flap extended beyond the deltopectoral groove, is too detrimental for tissue survival, especially in the distal portion, that thus undergoes necrosis. It is important to keep in mind that the distal portion beyond the deltopectoral groove can be classified as a random flap. Therefore, microvascular damage due to diabetes mellitus may have a greater impact in the distal part of the flap compared to the proximal portion, which is artery-based. Another risk factor could be a poor glycemic control during the postoperative period, with blood glucose levels higher than physiological values. This condition, combined with local tissue inflammation and the microvascular damage typical of diabetic subjects, could favor the onset of necrosis. A recent meta-analysis showed a statistically significant increase in flap failure in diabetic patients who underwent reconstructive surgery of the oral cavity, mainly including free flaps [28]. Although we did not find other studies reporting a percentage of total necrosis in patients with diabetes mellitus, one study suggested that diabetes mellitus, in addition to prior irradiation of the recipient site and wound infections, were the factors that contributed to flap necrosis, regardless of the use of the delay technique [29]. Concerning our results, the small number of subjects with diabetes mellitus was a limitation of the study. Indeed, other confounding factors may influence such a high complication rate in such a small number of subjects (n = 4/6).

Out of 31 patients, the only one with hypothyroidism experienced dehiscence of the surgical wound, and, subsequently, a flap necrosis. Although there is no statistically significant association, a larger sample size is needed for a more precise and accurate analysis of the occurrence of complications in hypothyroid subjects.

No increase in median hospital stay was observed in patients who developed postoperative complications. The median hospital stay in our study was 42 days (6 weeks), in agreement with the average hospital stay reported in the literature for patients who underwent reconstruction with a DP flap, which ranged from 6 to 10 weeks [26]. This result can be explained by a low total number of complications and the fact that the complications occurring in our patients mostly underwent a conservative treatment and did not prolong the hospital stay. On the contrary, a surgical intervention to solve the complication would have extended the hospital stay. It should also be noted that seven patients underwent surgery to close a pharyngocutaneous fistula that developed after total laryngectomy. These patients inevitably prolonged their hospital stay that began with the surgery for laryngeal cancer, without a DP flap in such procedure. It is interesting to observe that the age distribution of patients who experienced postoperative complications was similar to those who did not, suggesting that older age did not increase the risk of postoperative complications in DP flap. This was already highlighted in another study in 1980 [15].

Another interesting aspect is that the delayed technique has been recommended, starting from Bakamjan’s early studies, for patients with comorbidities such as diabetes mellitus, atherosclerosis, lupus erythematosus, malnutrition, anemia, or advanced age to reduce the occurrence of postoperative complications [30]. None of the 31 patients of our study underwent the delayed technique, indicating that complications were observed with the same frequency even in patients with comorbidity, except for diabetes mellitus.

## 5. Conclusions

The DP flap has a wide range of applications in head and neck reconstruction. Depending on the amount of tissue to be reconstructed, the distance from the donor site, and the patient’s characteristics, this flap can be adapted to achieve optimal reconstruction. The delayed technique does not necessarily need to be applied, and the flap can be extended beyond the deltopectoral groove when necessary. However, within the study group, although there was a small number of subjects, the patients with diabetes mellitus exhibited a higher risk of postoperative necrosis of the distal portion of the flap. Therefore, when dealing with diabetic subjects, the DP flap should be limited to the deltopectoral groove if possible. Further studies with larger samples and meta-analyses are mandatory to better evaluate the risk factors for postoperative complications in DP flaps.

## Figures and Tables

**Figure 1 medicina-60-01615-f001:**
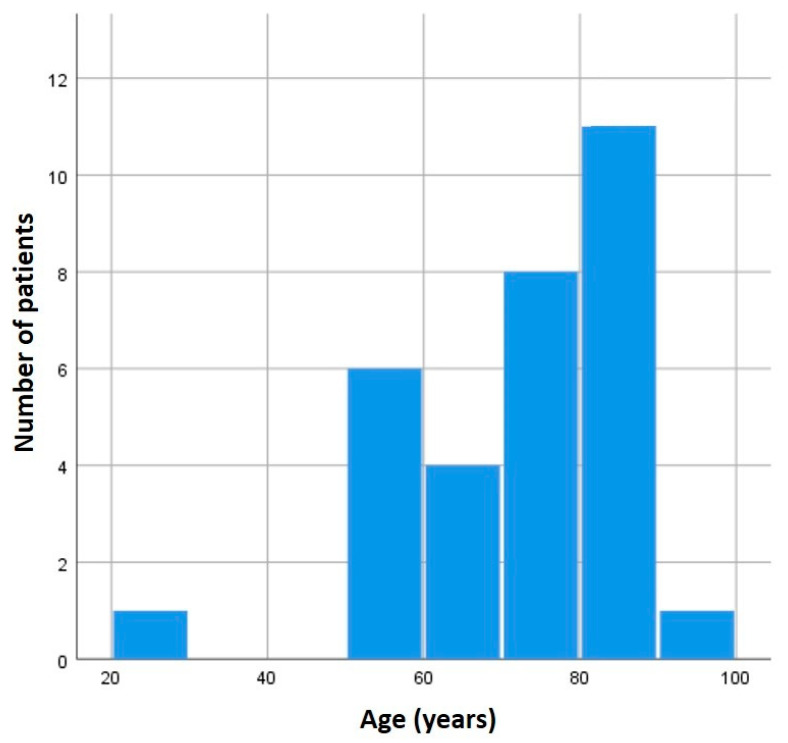
Distribution of age in the whole sample (*n* = 31).

**Figure 2 medicina-60-01615-f002:**
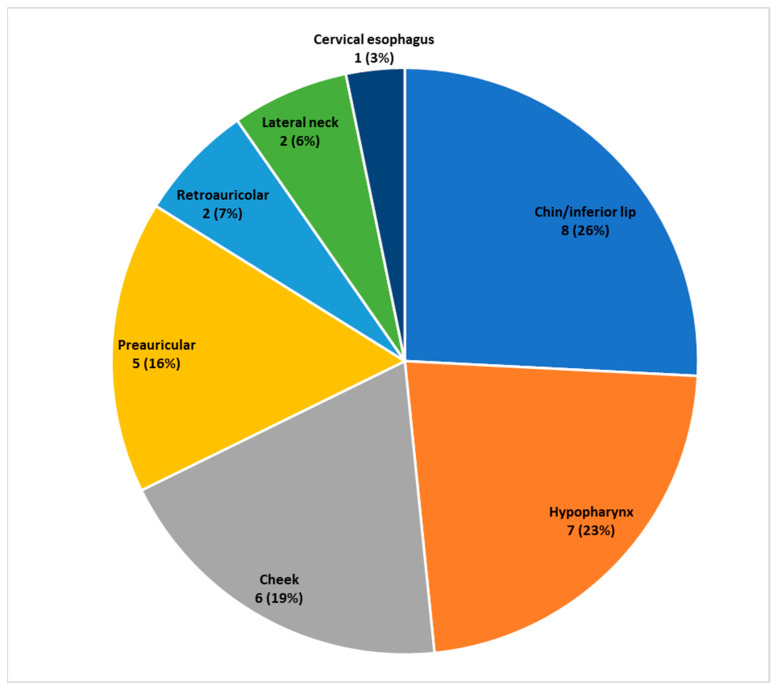
Recipient sites (*n*, %).

**Figure 3 medicina-60-01615-f003:**
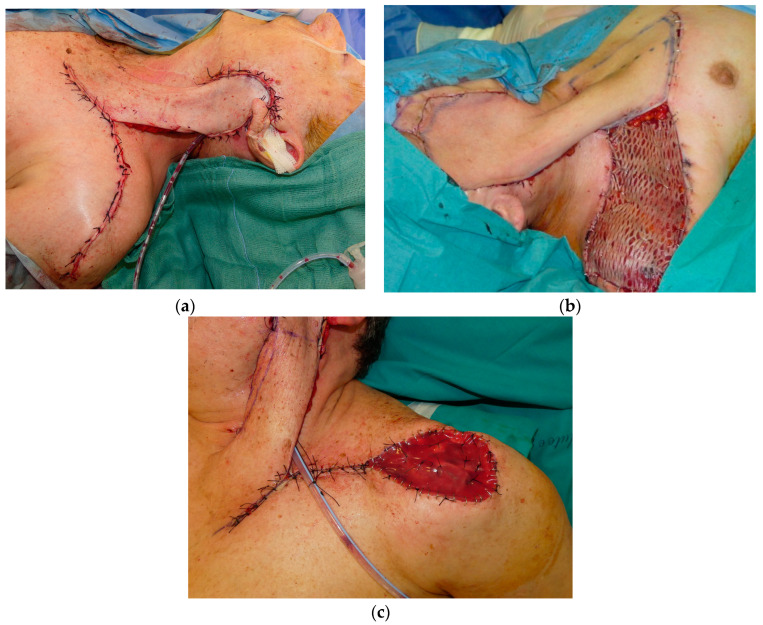
Closure of the donor site: (**a**) wide undermining and primary closure; (**b**) split-thickness skin autograft immediately after DP flap harvesting; (**c**) skin substitutes (bilayer Integra^®^ Dermal Regeneration Template, Integra Life Sciences, Plainsboro, NJ, USA) concomitant to DP flap harvesting and subsequent thin epidermal autograft concomitant to flap autonomization.

**Figure 4 medicina-60-01615-f004:**
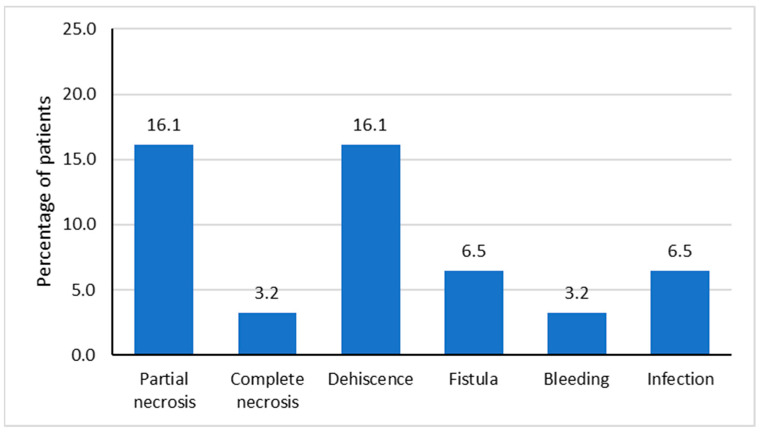
Postoperative complications.

**Table 1 medicina-60-01615-t001:** Clinical characteristics.

Characteristics	*N* (%)
Sex	
*Male*	19 (61.3)
*Female*	12 (38.7)
Smoking	
*Never*	21 (67.7)
*Former*	5 (16.1)
*Active*	5 (16.1)
Alcohol assumption	9 (29.0)
Previous head and neck radiotherapy	6 (19.4)
Diabetes mellitus	6 (19.4)
Systemic hypertension	13 (41.9)
Hypothyroidism	1 (3.2)
Chronic obstructive pulmonary disease	4 (12.9)
Liver cirrhosis	2 (6.5)
Chronic kidney disease	3 (9.7)

**Table 2 medicina-60-01615-t002:** Demolitive procedures before reconstruction with DP flap.

Procedure	*N* (%)
Maxillectomy	2 (6.4)
Parotidectomy extended to the skin	5 (16.1)
Segmental mandibulectomy	1 (3.2)
Neck dissection	1 (3.2)
Removal of skin carcinoma	14 (45.2)
Total laryngectomy	7 (22.6)
None (traumatic lesion of cervical esophagus)	1 (3.2)

**Table 3 medicina-60-01615-t003:** Correlations among clinical parameters and complications (p values at Mann–Whitney U-test for continuous variables, and chi-squared or Fisher exact test for categorical variables).

	Necrosis	Dehiscence	Any Complication
Age	0.095	0.735	0.159
Sex	0.653	0.948	0.373
Smoking	0.425	0.210	0.817
Alcohol assumption	0.796	0.627	0.935
Previous radiotherapy	0.853	0.968	0.363
Diabetes mellitus	0.006 *	0.553	0.067
Systemic hypertension	0.676	0.924	0.701
Hypothyroidism	0.194	0.161	0.323
COPD	0.759	0.525	0.577
Liver cirrhosis	0.474	0.521	0.313
Chronic kidney disease	0.372	0.424	0.533

COPD, chronic obstructive pulmonary disease. * *p* < 0.05.

**Table 4 medicina-60-01615-t004:** Complications reported in the literature and in our study.

Complications	Literature	Our Study
Partial necrosis	0–27% [14,24]	16.1%
Complete necrosis	0–26% [14,15]	3.2%
Dehiscence	0–21% [14,25]	16.1%
Fistula	3–18% [14,26]	6.5%
Bleeding	1% [14]	3.2%
Infection	0–27% [14,26]	6.5%

## Data Availability

The data presented in this study are available on request from the corresponding author.

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
