# Peer review of "Deltopectoral Flap in Head and Neck Reconstruction"

_medicina, 2024, doi:10.3390/medicina60101615_

Round 1

Reviewer 1 Report

Comments and Suggestions for Authors

1. Line 44: Please use the preferred term for the internal mammary artery (IMA) as defined in Terminologia Anatomica 2019, the "internal thoracic artery." Additionally, a more comprehensive description of the vessel's topography (and target organs of its supply) would be valuable.

2. Lines 71-72- Please provide more specific details about the studies reporting success rates, including the number of cases with the highest and lowest complication rates..

3. Methods: Since this is a retrospective study, it is unnecessary to mention the number of patients in the methods. Instead, consider relocating this information to the results. Begin the methods with a statement regarding ethical approval.

4. Lines 116-118: A citation is required for the statements: "The silicon layer provides a mechanical barrier and controls fluid loss. The matrix layer promotes cellular growth and collagen synthesis.."

5. Statistical Analysis: Please clarify the application of the Mann-Whitney U test by specifying the groups being compared.

6. Results-Provide additional information about the younger patient included in the study (aged 20-30 years), such as the indication for treatment. I  assumed that this patient that had a traumatic lesion?

7. Table 1. There is no need to abbreviate COPD within the table, as there appears to be sufficient space available.

8. Figure 2. Please correct the term "retroauricular". A spelling mistake..

9. Figure 4: Use decimal points (.) instead of commas (,) for decimal places to maintain consistency.

10. Discussion: While the discussion is well-written, it can be further enhanced by addressing another often-overlooked ENT flap covering similar field - sliding epiglottoplasty. It can be used  as a neopharyngeal reconstruction technique. This technique can be employed in patients with laryngeal or hypopharyngeal SCC who undergo total laryngectomy -Reference: https://doi.org/10.3389/fonc.2023.1284266

11. Conclusions: The conclusions section should emphasise the main findings of the research. Kindly remove the portion that mentions the original describer of the flap, as it does not contribute to the conclusions and has already been addressed twice earlier in the manuscript. Furthermore, restructure the sentence regarding the increased incidence of postoperative necrosis in diabetic patients. Consider rephrasing it to indicate that within the study group, patients with DM exhibited a higher rate of postoperative necrosis. Additionally, perhaps meta-analyses could provide even more detailed insights into the risk factors.

Comments on the Quality of English Language

Adequate

Author Response

  • Line 44: Please use the preferred term for the internal mammary artery (IMA) as defined in Terminologia Anatomica 2019, the "internal thoracic artery." Additionally, a more comprehensive description of the vessel's topography (and target organs of its supply) would be valuable.
  • Thanks for your comments and suggestions. The correct term and a description of the vessel totpography were added.

  • Lines 71-72- Please provide more specific details about the studies reporting success rates, including the number of cases with the highest and lowest complication rates.
  • More specific details were provided.

  • Methods: Since this is a retrospective study, it is unnecessary to mention the number of patients in the methods. Instead, consider relocating this information to the results. Begin the methods with a statement regarding ethical approval.
  • The number of patients was relocated in the Results. A statement regarding ethical approval is present in the first paragraph of the methods.

  • Lines 116-118: A citation is required for the statements: "The silicon layer provides a mechanical barrier and controls fluid loss. The matrix layer promotes cellular growth and collagen synthesis."
  • A citation was added.

  • Statistical Analysis: Please clarify the application of the Mann-Whitney U test by specifying the groups being compared.
  • We specified that the Mann-Whitney U test was used to compare subjects with or without complications.

  • Results-Provide additional information about the younger patient included in the study (aged 20-30 years), such as the indication for treatment. I assumed that this patient that had a traumatic lesion?
  • We specified that the younger patient had the traumatic lesion.

  • Table 1. There is no need to abbreviate COPD within the table, as there appears to be sufficient space available.
  • The abbreviation was removed.

  • Figure 2. Please correct the term "retroauricular". A spelling mistake.
  • The term “retroauricular” is currently used in literature.

  • Figure 4: Use decimal points (.) instead of commas (,) for decimal places to maintain consistency.
  • We corrected the error.

  • Discussion: While the discussion is well-written, it can be further enhanced by addressing another often-overlooked ENT flap covering similar field - sliding epiglottoplasty. It can be used as a neopharyngeal reconstruction technique. This technique can be employed in patients with laryngeal or hypopharyngeal SCC who undergo total laryngectomy -Reference: https://doi.org/10.3389/fonc.2023.1284266
  • Thank you for the suggestion. However, the sliding epiglottoplasty can be used for immediate pharyngeal reconstruction after total laryngectomy. The patients included in our study had a post-operative pharyngocutaneous fistula. Therefore, the sliding epiglottoplasty cannot be used in our cases.

  • Conclusions: The conclusions section should emphasise the main findings of the research. Kindly remove the portion that mentions the original describer of the flap, as it does not contribute to the conclusions and has already been addressed twice earlier in the manuscript. Furthermore, restructure the sentence regarding the increased incidence of postoperative necrosis in diabetic patients. Consider rephrasing it to indicate that within the study group, patients with DM exhibited a higher rate of postoperative necrosis. Additionally, perhaps meta-analyses could provide even more detailed insights into the risk factors.
  • The portion about the original describer of the flap was removed. The sentence regarding the necrosis in diabetic patients was restructured. Indication for meta-analysis was added.

Reviewer 2 Report

Comments and Suggestions for Authors

This retrospective study evaluates complications of deltopectoral flaps used for reconstructive surgery after head and neck surgeries.

While the authors should be congratulated for addressing a very timely topic with insufficient published literature, I believe there are limitations regarding the strength of the data.

Did the authors find any comparable data source showing a necrosis rate similar to 66% in patients with deltopectoral flaps and diabetes mellitus?

While the difference in complications between diabetic and non-diabetic patients is significant, other confounding factors may influence such a high complication rate in such a small number of subjects (n=4/6).

I believe this limitation would be important to address further in the discussion section and probably should not be the main point of this article's conclusion section.

Author Response

This retrospective study evaluates complications of deltopectoral flaps used for reconstructive surgery after head and neck surgeries.

While the authors should be congratulated for addressing a very timely topic with insufficient published literature, I believe there are limitations regarding the strength of the data.

Did the authors find any comparable data source showing a necrosis rate similar to 66% in patients with deltopectoral flaps and diabetes mellitus?

  • Thanks for your comments and suggestions. Although we did not find other studies reporting a percentage of total necrosis in patients with diabetes mellitus, one study suggested that diabetes, in addition to prior irradiation of the recipient site and wound infections, were the factors that contributed to flap necrosis, regardless of the use of the delay technique (reference n.26).

While the difference in complications between diabetic and non-diabetic patients is significant, other confounding factors may influence such a high complication rate in such a small number of subjects (n=4/6).

I believe this limitation would be important to address further in the discussion section and probably should not be the main point of this article's conclusion section.

  • The discussion of this limitation was added. The conclusions were rephrased according to your and Reviewer 1’s suggestions.

Reviewer 3 Report

Comments and Suggestions for Authors

This paper discusses the usefulness of the deltopectoral flap in head and neck reconstruction. 

The abstract is adequate, and has listed rationale and setting details, alongside the most important findings. The manuscript is written in fine English and does not need further editing.

The objectives of the study are presented clearly and the introduction section communicates the need for investigating the uses of the DP flap in head and neck reconstruction.

The main question is relevant and interesting and well described. The paper is well written and easy to read, and has no issues regarding internal or external validity.

The methods are valid, no power analysis has been performed due to a retrospective nature of the manuscript. IRB approval is noted. Adequate statistics have been used. 

However, novelty is a major problem, and the only significant problem with an otherwise well-crafted manuscript. The authors' patient cohort is underpowered for long-reaching conclusions, but well-described. The manuscript could be amended with an ancillary literature search, listing the most relevant recent papers concerning DP reconstruction in recent years and adding a paragraph listing the significance and novelty of this paper. This could improve the paper's usefulness and external validity. 

I would reconsider the paper after expansion to accommodate a short literature review.

Comments on the Quality of English Language

The language is fine, with minor points regarding wording and style. 

Author Response

This paper discusses the usefulness of the deltopectoral flap in head and neck reconstruction.

The abstract is adequate, and has listed rationale and setting details, alongside the most important findings. The manuscript is written in fine English and does not need further editing.

The objectives of the study are presented clearly and the introduction section communicates the need for investigating the uses of the DP flap in head and neck reconstruction.

The main question is relevant and interesting and well described. The paper is well written and easy to read, and has no issues regarding internal or external validity.

The methods are valid, no power analysis has been performed due to a retrospective nature of the manuscript. IRB approval is noted. Adequate statistics have been used.

However, novelty is a major problem, and the only significant problem with an otherwise well-crafted manuscript. The authors' patient cohort is underpowered for long-reaching conclusions, but well-described. The manuscript could be amended with an ancillary literature search, listing the most relevant recent papers concerning DP reconstruction in recent years and adding a paragraph listing the significance and novelty of this paper. This could improve the paper's usefulness and external validity.

  • Thanks for your comments and suggestions. The most recent relevant papers were added and the novelty of this paper was explained in the second paragraph of the discussion.

Round 2

Reviewer 1 Report

Comments and Suggestions for Authors

The authors addressed the queries and suggestions. No further comments.

Comments on the Quality of English Language

Adequate

Reviewer 2 Report

Comments and Suggestions for Authors

Thank you for addressing my concerns. I have no further comments.

Reviewer 3 Report

Comments and Suggestions for Authors

The authors have responded to the ciriticism raised in the initial review and have improved the quality of the manuscript. 

Comments on the Quality of English Language

No major issues.